# Angiogenetic Factors in Hepatocellular Carcinoma During Transarterial Chemoembolization: A Pilot Study

**DOI:** 10.3390/cancers17162642

**Published:** 2025-08-13

**Authors:** Joško Osredkar, Špela Koršič, Uršula Prosenc Zmrzljak, Hana Trček, Peter Popović

**Affiliations:** 1Institute of Clinical Chemistry and Biochemistry, University Medical Centre Ljubljana, Zaloška 2, 1000 Ljubljana, Slovenia; 2Faculty of Pharmacy, University of Ljubljana, Aškerčeva Cesta 7, 1000 Ljubljana, Slovenia; 3Clinical Institute of Radiology, University Medical Centre Ljubljana, Zaloška 2, 1000 Ljubljana, Slovenia; spela.korsic@kclj.si (Š.K.);; 4Faculty of Medicine, University of Ljubljana, Vrazov trg 2, 1000 Ljubljana, Slovenia; 5Molecular Biology Laboratory, BIA Separations CRO, Labena Ltd., 1000 Ljubljana, Slovenia; 6Centre for Functional Genomics and Bio-Chips, Institute of Biochemistry and Molecular Genetics, Faculty of Medicine, University of Ljubljana, Zaloška Cesta 4, 1000 Ljubljana, Slovenia

**Keywords:** hepatocellular carcinoma, transarterial chemoembolization, angiogenesis, angiopoietin-2, HGF, VEGF-A, FGF-1, FGF-2, endothelin-1

## Abstract

Transarterial chemoembolization (TACE) is the treatment of choice for the intermediate-stage hepatocellular carcinoma (HCC). While TACE effectively induces tumor necrosis through ischemia, hypoxia-triggered angiogenesis may contribute to HCC recurrence. This study evaluated plasma angiogenic factors in 25 HCC patients pre- and post-TACE. Angiopoietin-2 peaked at three days post-TACE before declining at one month, while hepatocyte growth factor spiked at 24 h and later normalized. Endothelin-1 showed a temporary increase in four patients. Fibroblast growth factors (1 and 2) and vascular endothelial growth factor A were rarely detected, potentially reflecting either minimal systemic release or assay sensitivity issues. These findings demonstrate that TACE provokes transient angiogenic responses, likely secondary to ischemia. Further studies should refine detection methods and explore the utility of these factors as prognostic markers or therapeutic targets to improve HCC management.

## 1. Introduction

According to the Barcelona Clinic Liver Cancer (BCLC) staging system, transarterial chemoembolization (TACE) is the treatment of choice for intermediate-stage (BCLC-B) hepatocellular carcinoma (HCC) [1,2]. Despite its widespread use, TACE is not a standardized procedure; both the technique and treatment strategies vary between institutions. TACE is performed using the so-called conventional method (conventional TACE, cTACE) or the more recently established method involving drug-eluting microspheres (DEM-TACE) [1,3]. In cTACE, the chemotherapeutic agent is administered in a lipiodol emulsion, whose properties enable selective uptake and retention within the highly vascularized tumors characteristic of HCC [4]. Compared to cTACE, the improved pharmacokinetic profile of DEM-TACE enables the slow and controlled release of the chemotherapeutic agent, resulting in a strong cytotoxic and ischemic effect at the target tumor site, while also reducing systemic side effects [5]. However, no significant difference in overall survival has been observed between the two techniques [6].

TACE demonstrates therapeutic utility across various HCC stages, from early to advanced disease. In early-stage HCC (BCLC-A), it is used as a bridging therapy to liver transplantation by maintaining tumor control, or in patients who are not candidates for percutaneous ablation, resection or liver transplantation. In advanced stages (BCLC-C), it may be combined with systemic agents such as sorafenib or lenvatinib. Furthermore, TACE shows synergistic effects when used combined with locoregional therapies including percutaneous ablation, potentially enhancing treatment outcomes through complementary mechanisms of action [2,6].

Idarubicin, a lipophilic anthracycline, has shown promise in the treatment of HCC through TACE [7,8,9,10]. Its superior cytotoxicity, demonstrated in vitro on three HCC cell lines of viral origin, along with its capacity to overcome multidrug resistance, makes it a promising alternative to doxorubicin, which remains the most used chemotherapeutic agent in TACE to date [11,12].

TACE involves selective catheterization of the tumor feeding artery and local delivery of chemotherapeutic agents that are either emulsified with lipiodol in cTACE, or loaded to microspheres in DEM-TACE, creating a hypoxic and cytotoxic environment. However, the resultant hypoxia is a potent trigger for pro-angiogenic signaling pathways, particularly through the upregulation of hypoxia-inducible factors (HIFs) and downstream mediators such as vascular endothelial growth factor (VEGF) [13,14]. This compensatory angiogenesis can paradoxically promote tumor survival, recurrence, and even metastasis by restoring blood flow and facilitating dissemination of tumor cells. The newly formed tumor vasculature is often structurally abnormal (damaged endothelium) and functionally inefficient, which contributes to persistent hypoxia and creates a so-called vicious cycle that favors aggressive tumor behavior (i.e., promotes tumor recurrence and metastasis) [13,15,16].

Combining TACE with anti-angiogenic therapies (e.g., sorafenib, bevacizumab) or immunotherapy is a promising strategy to enhancing treatment outcomes. This strategy may produce synergistic or abscopal effect, potentially suppressing tumor regrowth and reducing the risk of metastasis [17,18,19].

To elucidate the dynamic changes in angiogenic signaling and their potential impact on therapeutic outcomes, we evaluated key angiogenic parameters before and during TACE. We aimed to characterize temporal alterations of vascular remodeling and identify potential biomarkers predictive of treatment response. We wanted to understand the relationship between tumor ischemia, vascular remodeling, and possible resistance mechanisms after idarubicin-loaded DEM-TACE by examining circulation levels of angiopoietin-2, hepatocyte growth factor (HGF), endothelin-1, fibroblast growth factors 1 and 2 (FGF-1 and FGF-2), and VEGFs. Although VEGF-A is the most extensively studied pro-angiogenic factor, other isoforms such as VEGF-C and VEGF-D also play significant roles, particularly in lymphangiogenesis and metastatic spread. Elevated expression of VEGF-C and VEGF-D in HCC has been associated with lymph node metastasis and poor prognosis [20]. Thus, incorporating these isoforms into the biomarker panel could provide a more comprehensive understanding of lymphatic and vascular responses following embolization.

Recent advances in the molecular classification of HCC highlight distinct angiogenic and immune phenotypes that influence treatment response and biomarker development [21]. Moreover, biomarker-guided strategies are increasingly being explored to optimize therapeutic outcomes and personalize treatment in HCC patients [22].

## 2. Materials and Methods

### 2.1. Patients

A total of 25 patients with biopsy-confirmed intermediate-stage HCC who underwent idarubicin-loaded DEM-TACE were included in this study. Baseline clinical data were collected for all 25 patients, including presence of cirrhosis, etiology of liver disease, Child–Pugh class, as well as lesion count and diameter of the biggest lesion. Most patients (96%), except one, presented with cirrhotic liver, of them 66.7% of ethylic etiology. Less common causes of cirrhosis were non-alcoholic steatohepatitis (16.7%), hemochromatosis (8.3%), and hepatitis C infection (4.2%). In one patient, no underlaying cause of cirrhosis was identified. Cirrhosis was classified as Child–Pugh A and B in 19 (79.2%) and 5 (20.8%) patients, respectively. Mean number of lesions was 3.2 ± 2.2 mm and mean diameter of the largest lesion was 46.8 ± 23.6 mm. Four patients had previously undergone TACE. Informed consent was obtained from all patients. Peripheral blood samples were collected at baseline (pre-TACE), 24 h, 3 days, and 1 month post-TACE. Plasma levels of angiogenetic factors were analyzed using a multiplex bead-based assay. Statistical analysis was performed to assess temporal changes in factor levels.

### 2.2. Method

Plasma samples were stored at −70 °C until the analysis. On the day of the analysis, the samples were thawed on ice, vortexed, and briefly spun. All samples from the same patient were analyzed on the same plate, with each sample in technical replicate. The following analytes were measured: angiopoietin-2, EGF, endoglin, endothelin-1, FGF-1, FGF-2, HGF, VEGF-A, VEGF-C and VEGF-D with milliplex human angiogenesis/growth factor magnetic bead panel (Merck Millipore, Darmstadt, Germany, HAGP1MAG-12K), following the producer’s protocol. The readout was performed on Bio-Plex 200 System (Bio-Rad, Hercules, CA, USA) with program Bio-Plex Manager version 6.2 (Bio-Rad). Analyte concentration in each sample was calculated using software from standard curve that was included in each plate.

### 2.3. Statistical Analysis

First, normality of the collected data was tested with Shapiro–Wilk and Kolmogorov–Smirnov tests. Since all the obtained data did not follow normal distribution, we used a nonparametric Friedman test to compare plasma concentrations of angiogenesis factors measured at four time points (pre-TACE, 24 h, 72 h and 1 month post-TACE). Values reported as OOR (out of range) indicate analyte concentrations below the assay’s lower limit of quantification (LLOQ). In statistical analyses, these were treated as non-detectable. For visualization purposes (e.g., Section 3.1), OOR values were imputed as zero solely for graphical representation. Multiple comparisons were assessed using Dunn’s test, and significance was considered at an adjusted *p*-value ≤ 0.05. All statistical analyses were performed using GraphPad Prism 10.2.0.

## 3. Results

A similar trend is seen across all measured angiogenesis factors (Table 1 and Figure 1). Plasma levels of angiopoietin-2 were significantly elevated 3 days post-TACE, with decreasing levels at the 1-month follow-up. A similar pattern was observed with HGF, where significantly elevated plasma levels were detected 24 h and 72 h post-TACE, followed by a decrease after 1 month. Endothelin-1 also showed a transient increase in plasma, followed by a decrease; however, it was detected in only four patients. Also, FGF-1, FGF-2, and VEGF-A factors were detected in a limited number of patients (n = 7, n = 9, n = 4, respectively), making statistical analysis challenging.

Table 1 summarizes absolute plasma concentrations, while Figure 1 displays the time-course dynamics for each factor. To enable cross-comparison, Figure 2 shows relative values normalized to baseline, highlighting distinct temporal patterns in biomarker response.

### 3.1. Relative Changes in Angiogenesis Biomarkers

The plasma levels of important angiogenic factors were adjusted to their respective pre-TACE values (set at 100%) to more easily evaluate trends among biomarkers with varying baseline concentrations. Figure 2 provides a clear illustration of the relative biomarker kinetics, facilitating cross-marker interpretation and revealing a biphasic response pattern:•At 72 h after TACE, angiopoietin-2, HGF, and endothelin-1 all exhibited significant increases, reaching 123%, 148%, and 224% of their baseline values, respectively, before beginning to decline one month later.•In contrast, VEGF-A and EGF levels were significantly reduced within 24 h (81% and 0%, respectively). While EGF remained suppressed at 72 h (7%) with a partial recovery at 1 month (61%), VEGF-A showed a rebound at 72 h (116%) followed by a fall at 1 month (46%).

These results show a rebound response compatible with hypoxia-driven angiogenesis and tissue remodeling after a brief early reduction in pro-angiogenic signals immediately following TACE.

**Figure 2 cancers-17-02642-f002:**
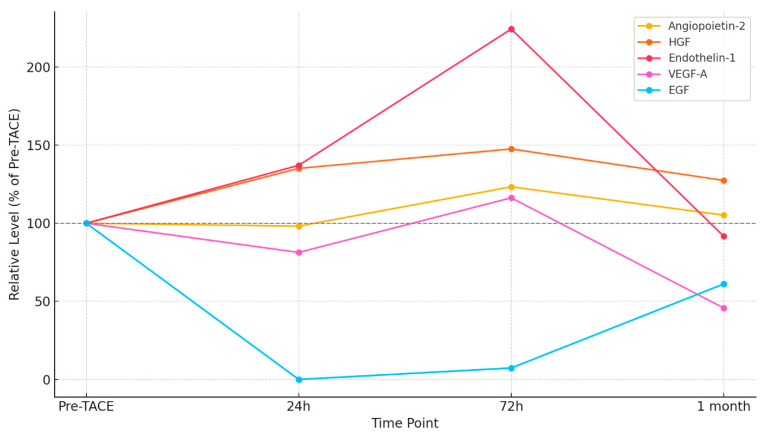
Relative changes in angiogenic biomarkers after TACE, normalized to pre-TACE baseline (100%). Five biomarkers—angiopoietin-2, HGF, endothelin-1, VEGF-A, and EGF—are shown across four time points: baseline, 24 h, 72 h, and 1 month post-TACE.

### 3.2. Time-Dependent Relative Changes

To better compare trends across biomarkers with different baseline concentrations, plasma levels were normalized to their respective pre-TACE values.

Angiopoietin-2 showed a significant increase at 72 h post-TACE, consistent with previous reports linking its elevation to endothelial destabilization and vascular remodeling. The transient peak and subsequent decline suggest a temporally restricted hypoxia response, which may correspond to endothelial stress during vascular collapse and early remodeling.

HGF increased significantly at both 24 h and 72 h, reflecting hepatocyte response to ischemic injury. This increase is consistent with hepatocyte regenerative signaling and could indicate not only tumor escape signaling but also liver tissue repair dynamics.

Although detected in only four patients, endothelin-1 showed the strongest relative increase (224% at 72 h), highlighting its potential role in the early vasoconstrictive phase post-TACE. Its rapid normalization may reflect tight regulatory control or localized effects.

VEGF-A rebounded at 72 h, aligning with HIF-1α mediated angiogenesis post-hypoxia. VEGF-C and VEGF-D showed partial recovery, but with significant interindividual variability, which may relate to differences in lymphangiogenic activity or tumor phenotype.

EGF was completely suppressed at 24 h and showed delayed, partial recovery at 1 month, possibly reflecting lasting suppression of epithelial proliferative signaling. Its behavior differs markedly from the other factors, suggesting distinct temporal regulation.

### 3.3. Preliminary Clinical Observations and Biomarker Interpretation

Although this study was not designed to correlate biomarker changes with radiological or survival outcomes, preliminary follow-up data were consistent with biochemical trends. Specifically, imaging performed one month after the first cycle of TACE—which, depending on disease burden, consisted of one to three treatments—revealed reduced vascularization on contrast-enhanced computed tomography in 92% of patients (23/25), consistent with the observed decrease in VEGF-A and angiopoietin-2 levels at the same time point.

Moreover, the marked elevation in HGF and endothelin-1 at 72 h post-procedure may reflect hepatocyte stress and vascular remodeling, mechanisms that could contribute to post-TACE escape or recurrence. These early shifts in angiogenic signaling may serve as surrogate indicators of tumor behavior and therapeutic response, highlighting their translational potential in clinical monitoring and stratification.

Several angiogenic factors (e.g., FGF-1, VEGF-A) showed low detection rates or out-of-range (OOR) values, which may reflect true low systemic concentrations or limitations of multiplex assay sensitivity. This limitation is acknowledged and motivates the use of ultrasensitive platforms in follow-up studies.

### 3.4. Summary of Biomarker Response Patterns

The combined analysis of absolute and normalized values reveals a clear biphasic pattern in the angiogenic response following TACE. An early suppression phase is evident for VEGF-A and EGF at 24 h, likely reflecting the acute ischemic effect of embolization. This is followed by a rebound phase at 72 h for angiopoietin-2, HGF, and endothelin-1, highlighting a delayed but coordinated pro-angiogenic and endothelial activation response.

This temporal pattern emphasizes the biological impact of TACE beyond immediate tumor necrosis. The observed changes in circulating biomarkers suggest that the tumor microenvironment undergoes rapid adaptation involving both angiogenic reactivation and potential mechanisms of escape. While individual markers exhibit distinct kinetics, the synchronized rise in angiopoietin-2, HGF, and endothelin-1 around 72 h may represent a critical window for intervention with adjunct therapies.

Moreover, the decline in most markers by 1 month suggests partial resolution of this response, though longer follow-up would be needed to determine the clinical trajectory. These findings support the value of dynamic biomarker profiling in understanding and potentially guiding TACE-based treatment strategies.

## 4. Discussion

The observed transient increase in angiopoietin-2, HGF, and endothelin-1 following TACE aligns with previous reports and likely reflects hypoxia-induced angiogenic responses [13]. Several studies have investigated the role of angiogenic factors in HCC and their dynamic changes post-TACE, highlighting their potential as biomarkers for tumor response, progression, and therapeutic resistance [23,24,25].

### 4.1. Angiopoietin-2 and Tumor Vascular Remodeling

Angiopoietin-2 plays a critical role in vascular remodeling, endothelial destabilization, and angiogenic escape following hypoxic stress. Elevated circulating levels post-TACE have been widely reported [26,27]. Choi et al. demonstrated that plasma levels of angiopoetin-2 significantly increase with disease progression and show superior predictive power for overall survival and progression-free survival compared to other angiogenic markers like angiopoetin-1 and VEGF [28]. Similarly, Balli et al. found that a rapid rise in angiopoetin-2 correlated with increased microvascular density in post-TACE tumor biopsies, supporting the hypothesis that TACE-induced hypoxia enhances angiogenic escape mechanisms [29]. These findings highlight the complex role of angiopoetin-2 in tumor angiogenesis and its potential as a therapeutic target in cancer treatment.

### 4.2. HGF and Tumor Cell Survival

Hepatocyte growth factor (HGF) is a key mitogen involved in liver regeneration, tumor cell survival, and metastasis [30]. Studies have shown transient HGF elevations post-TACE, likely due to ischemia-induced hepatocyte injury and stromal activation [31]. The HGF/c-MET signaling pathway plays a crucial role in cell proliferation, migration, invasion, and survival, with its dysregulation implicated in various cancers, including hepatocellular carcinoma (HCC) [32,33].

### 4.3. Endothelin-1 and Hypoxia Adaptation

Endothelin-1 (ET-1), a vasoconstrictive peptide, is known to contribute to hypoxia-driven vascular remodeling [34]. Karimi et al. demonstrated that endothelin-1 expression increased in hypoxic regions of HCC following TACE, with elevated levels predicting a higher likelihood of tumor recurrence [35]. In another study, Li H et al. found that inhibiting endothelin-1 enhanced the efficacy of anti-angiogenic therapies post-TACE, suggesting its potential as a therapeutic target [36].

### 4.4. Limited Detection of FGF-1, FGF-2, and VEGF-A

Fibroblast growth factors (FGF-1, FGF-2) and vascular endothelial growth factor A (VEGF-A) are well-characterized angiogenic mediators, yet their plasma levels post-TACE were detected in only a few (4/25) patients in our study. Possible explanations include rapid clearance, localized tumor release, or insufficient assay sensitivity.

### 4.5. VEGF-C and VEGF-D

VEGF-C and VEGF-D are primarily involved in lymphangiogenesis and have been associated with extrahepatic spread [37]. In our study, their levels decreased 24 h after TACE but subsequently showed partial recovery. However, due to substantial interindividual variability, these changes did not reach statistical significance. While the precise role of VEGF-C and VEGF-D in HCC progression post-TACE remains unclear, these observations may suggest transient suppression of lymphatic signaling following embolization. Further targeted studies are needed to determine their prognostic relevance.

### 4.6. Relative Changes in Angiogenesis Biomarkers

For inter-marker comparability, we normalized post-treatment values to baseline levels, allowing clearer identification of temporal kinetics regardless of absolute concentration variability. This approach maintained consistency with observed trends in raw data.

Figure 2 demonstrates temporal trends by highlighting the relative changes in biomarker levels by normalizing them to pre-TACE baselines. Their function as early hypoxia-responsive mediators is further supported by the notable rise in endothelin-1 (224%) and HGF (148%) at 72 h. A delayed, hypoxia-induced angiogenic surge is further supported by the recovery of VEGF-A following initial suppression; this pattern may represent tumor escape mechanisms following embolization.

Remarkably, EGF was not detected at 24 h and only partially recovered at 1 month, which could indicate stromal remodeling or longer-term suppression of epithelial growth. EGF differs from the quick but fleeting reactions of traditional angiogenic indicators such as VEGF and angiopoietin-2 due to its delayed response.

Although we observed patterns consistent with hypoxia-driven angiogenesis, our study did not directly measure HIF-1α or VEGF expression in tissue. These markers are more suitable for biopsy-based studies or immunohistochemical analysis. Future research incorporating tissue-level evaluation could better link circulating factors to hypoxia-related signaling.

### 4.7. Clinical Implications and Future Directions

The transient increase in angiogenic factors following TACE suggests a therapeutic window for targeted intervention. Combination therapies integrating TACE with anti-angiogenic agents (e.g., sorafenib, lenvatinib, bevacizumab) have shown promise in prolonging tumor control [38].

To enhance biomarker utility, future studies should optimize detection methods using ultrasensitive ELISA, single-molecule array (Simoa), or mass spectrometry. In future studies, plasma biomarkers could be complemented with dynamic imaging (e.g., perfusion CT or DCE-MRI) or tissue radioligand binding assays. Comparing responses between DEM-TACE and transarterial radioembolization (TARE) could further clarify treatment-specific angiogenic effects.

Additionally, the biological relevance of these angiogenic factors in the context of TACE should be further investigated. Angiopoietin-2 has been implicated in promoting vascular remodeling and tumor adaptation to hypoxia, suggesting that its transient increase post-TACE may contribute to tumor progression. Similarly, HGF is a potent mitogen that has been linked to tumor cell survival and migration, raising the possibility that its upregulation following TACE may facilitate residual tumor growth. The role of endothelin-1 in tumor microcirculation and hypoxia adaptation also warrants further investigation, as its transient increase may have implications for vascular resistance and perfusion changes post-TACE.

Given the heterogeneous responses observed, patient stratification based on angiogenic profiles may provide insights into personalized therapeutic approaches. Identifying subgroups of patients who exhibit distinct angiogenic responses could lead to tailored interventions, such as combining TACE with anti-angiogenic therapies to mitigate hypoxia-induced tumor escape mechanisms.

Future studies should also consider incorporating imaging biomarkers (e.g., dynamic contrast-enhanced MRI and perfusion CT) alongside circulating angiogenic factors to better understand the spatial and temporal dynamics of tumor vascularization post-TACE. This integrated approach may refine treatment response prediction and guide personalized therapeutic strategies.

We also acknowledge the lack of functional validation of angiogenic activity. While Ang-2 and HGF were elevated, we did not assess their in vitro activity on endothelial or tumor cells. This would be an important next step for future mechanistic work. In addition, we did not evaluate mRNA expression or tumor biopsies, which could clarify the source of systemic biomarkers.

### 4.8. Methodological Considerations and Study Limitations

This study offers several methodological strengths. The use of four distinct sampling time points allowed us to characterize both immediate and delayed biomarker responses following TACE. By analyzing samples at 24 h, 72 h, and 1 month in comparison to baseline, we captured the biphasic nature of angiogenic signaling and potential windows for therapeutic intervention. Additionally, the application of a multiplex bead-based assay enabled parallel quantification of multiple angiogenic factors from small plasma volumes, ensuring assay consistency and allowing direct temporal comparison across markers.

Normalization of values to pre-TACE baseline levels further improved the interpretability of dynamic changes, particularly for markers with different absolute ranges. This approach highlighted key rebound responses that might be less evident using absolute concentrations alone.

However, some limitations must be acknowledged. The sample size was limited to 25 patients, consistent with a pilot design, which may constrain the generalizability and statistical power of our findings. Certain biomarkers, such as endothelin-1 and EGF, were detectable in only a subset of patients or fell below the assay’s detection threshold (reported as OOR), which may limit the robustness of conclusions drawn from those markers. Furthermore, although the study captured biological dynamics, clinical outcomes such as radiological response or survival were not formally correlated with biomarker trajectories. Future studies incorporating larger cohorts and clinical endpoints are warranted to validate and extend these findings.

Finally, our sampling period was limited to one month. Since some reports suggest a secondary angiogenic peak at 2–3 months post-TACE, extending follow-up time points in future studies will provide a more complete view of angiogenic rebound [14,39,40].

## 5. Conclusions

Our findings suggest that TACE induces a transient but coordinated angiogenic response, characterized by increased levels of angiopoietin-2 and HGF, particularly at 72 h post-treatment. The role of other angiogenic factors, such as endothelin-1, VEGF isoforms, and EGF, remains less clear due to variability and detection limitations. Further research is needed to validate these biomarkers in larger cohorts and determine their clinical significance in predicting treatment outcomes, stratifying patients, and guiding therapeutic interventions. Incorporating more sensitive detection platforms and integrating biomarker dynamics with imaging-based assessments may enhance the precision of HCC treatment monitoring and decision-making.

## Figures and Tables

**Figure 1 cancers-17-02642-f001:**
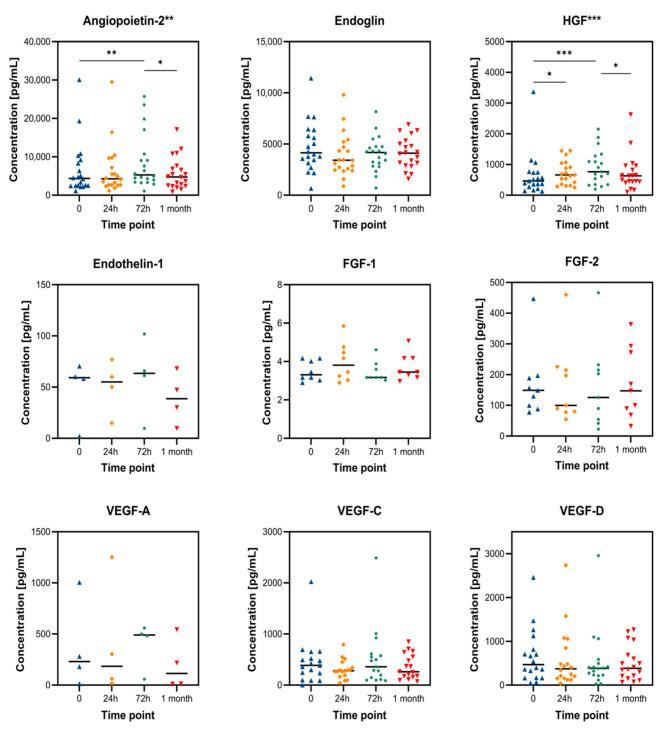
Changes in plasma concentration of angiogenesis factors after TACE therapy. Statistical significance from the Friedan test is indicated beside the factor name, while Dunn’s test significance is shown within the graph, where * ≤0.05, ** ≤0.01, and *** ≤0.001. Data are represented as mean ± SD.

**Table 1 cancers-17-02642-t001:** Concentrations of measured angiogenesis factors in 25 HCC patients who underwent TACE therapy [pg/mL].

	pre-TACE	24 h After TACE	72 h After TACE	1 Month After TACE
Angiopoietin-2	4524.2 ± 6291.30	4439.3 ± 5941.87	5578.7 ± 6869.23	4758.4 ± 4026.21
Endoglin	4011.85 ± 2232.37	3406.4 ± 1947.87	4074.42 ± 1585.61	3879.17 ± 1472.43
HGF	500.27 ± 679.12	675.87 ± 411.02	738.08 ± 539.76	637.33 ± 590.04
Endothelin-1	27.28 ± 26.56	37.39 ± 25.82	61.19 ± 36.29	24.99 ± 23.74
FGF-1	3.31 ± 284.76	4.17 ± 257.88	3.17 ± 84.81	3.45 ± 54.21
FGF-2	143.24 ± 96.20	106.72 ± 109.03	143.84 ± 148.88	118.54 ± 96.24
VEGF-A	78.1 ± 285.49	63.5 ± 363.20	90.79 ± 191.80	35.7 ± 184.89
VEGF-C	363.31 ± 400.61	268.03 ± 177.47	283.06 ± 529.75	248.69 ± 239.77
VEGF-D	363.5 ± 582.91	255.75 ± 650.69	319.28 ± 629.35	383.07 ± 399.41
EGF	98.3	OOR<	7.19 ± 33.15	60.1 ± 9.33

Data are represented as mean ± SD. SD, standard deviation.

## Data Availability

The data presented in this study are available in this article.

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
