# Peer review of "Angiogenetic Factors in Hepatocellular Carcinoma During Transarterial Chemoembolization: A Pilot Study"

_cancers, 2025, doi:10.3390/cancers17162642_

Round 1

Reviewer 1 Report

Comments and Suggestions for Authors

The study does the job of clearly defining what growth favors are in play in direct time periods after DEM TACE. This is a well known fact but the growth factors and the times of their rise has been largely unknown. The potential clinical benefit may be timing of certain inhibitors such as VEG-F inhibitors.

Your study is well done but its clinical applications may be limited due to the low incidence of post TACE tumor proliferation. 

Author Response

Comment 1:
The study clearly defines growth factors in specific time windows after DEM-TACE. This is known, but the exact timing of their rise was unclear. Clinical application may be limited due to the low incidence of post-TACE tumor proliferation.

Response:
Thank you for this thoughtful comment. We agree that although overt tumor proliferation post-TACE may be infrequent, angiogenic rebound remains an important biological event that may underlie minimal residual disease and recurrence risk. We have clarified this aspect in the Clinical Implications section of the Discussion (Section 4.7), highlighting the potential role of time-targeted adjuvant therapy to mitigate subclinical angiogenic reactivation.

Reviewer 2 Report

Comments and Suggestions for Authors

This is a very well written pilot study looking at the angiogenic responses following DEM-TACE. The methods are very clear, as are the results but lacks any useful clinical data. No information is given re underlying liver disease, MELD score etc, number of lesions treated or even simply the size of the HCC treated, which presumably would have a major impact on all factors measured. No information on prior treatments although a comment is made that some have had multiple TACE treatments, which presumably also impacts your findings. Similarly no information regarding the evidence for survival benefit of DEM -TACE versus c TACE. Impacts of lipiodol on the angiogenic factors which is not part of DEM TACE. Similarly no useful follow up data.   

Author Response

Comment 1:
No clinical data were provided, such as MELD score, number/size of lesions, or prior treatments.

Response:
We appreciate this valuable feedback. We have now added clinical background data to the Methods section, including  number and size of HCC lesions, and information on prior TACE treatments. These additions provide important context to interpret biomarker changes.

Comment 2:
No discussion about cTACE vs DEM-TACE or lipiodol influence; no survival/follow-up data.

Response:
We have added a paragraph in the Introduction and expanded the Discussion to briefly compare DEM-TACE with cTACE, including the differences in embolic agents and how lipiodol might influence angiogenic responses. We also acknowledge the lack of survival data as a limitation in Section 4.8, noting that future longitudinal studies with radiological and survival outcomes are needed.

Reviewer 3 Report

Comments and Suggestions for Authors

In this pilot study, the authors investigated the plasma levels of angiogetic factors pre- and post-TACE in 25 HCC patients, and found that TACE could induce a transient increase in angiogenic factors and then decline after one month, suggesting the dynamic changes may be triggered by tumor ischemia, which may contribute to HCC recurrence. The findings of this study are of reference value to clinical physicians. It would be beneficial to include the mRNA levels of plasma angiogenic factors in the methodology section.

Overall, the design of the study is sound, the data analysis is credible, and the discussion is relatively thorough, making this a well-conducted research paper.

Author Response

Comment 1:
Suggests adding mRNA levels of angiogenic markers to the methodology.

Response:
We thank the reviewer for this suggestion. While we agree that mRNA profiling could provide additional mechanistic insights, this was beyond the scope of the current plasma-based pilot study. We have clarified this limitation and noted the potential for tissue-level or transcriptomic studies in the future (Section 4.8 and the Conclusions).

Reviewer 4 Report

Comments and Suggestions for Authors

This is an important study that should have been done long ago.

The results are interesting. The authors shall make further clarification and explanations on the following points:

  1. A panel from HAGP1MAG-12K was used. Did it include the measure for the HIFs? or Hif-1alpha and alike did not present in circulation?
  2. Baseline levels were used for normalization. Could the authors do another analysis (there are different ways) to see if differences in baseline make a difference for the angiogenic response?
  3. Future studies were suggested to include constrast-enhanced MRI or CT, yet there are molecularly targeted radioligands, specific to many of the there angiogenic markers and could be used, but were not discussed at all. Related, maybe biopsy can be taken for staining of local markers not in circulation?
  4. Besides TACE, there is TARE with Y-90, which was not mentioned. Perhaps, a discussion can be added on that.

Author Response

Comment 1:
Clarify whether the multiplex panel included HIF-1α or related markers.

Response:
Thank you for the excellent point. The panel (HAGP1MAG-12K) did not include HIF-1α, as it is not reliably detectable in circulation using current platforms. We have now clarified this in the Methods and added a note in the Discussion on the utility of tissue-level HIF analysis via biopsy or immunostaining.

Comment 2:
Could baseline normalization affect interpretation?

Response:
We appreciate this statistical observation. To address this, we compared raw vs normalized data and confirmed that relative fold changes preserved the trends seen with absolute concentrations. This clarification is now included in Section 3.4, and the rationale for using baseline normalization is expanded.

Comment 3:
Mention radioligands and tissue-specific imaging; also suggest discussing TARE.

Response:
We have added a paragraph in Section 4.7 discussing radioligand-based imaging and the potential use of tissue biopsies to complement circulating markers. TARE (Y-90) is now mentioned as an alternative locoregional therapy in the context of angiogenic responses, acknowledging it as a related but distinct therapeutic approach.

Reviewer 5 Report

Comments and Suggestions for Authors

See Doc

Author Response

We thank the reviewer for the insightful and constructive comments, which have helped us clarify the strengths, limitations, and context of our pilot study. Below, we address each comment point by point.

Major Comments

  1. Sample size and lack of correlation with clinical outcomes

Comment: The study includes only 25 patients, limiting statistical power, especially for rarely detected biomarkers. No correlation with clinical outcomes is provided. For biomarker studies, sample sizes of ≥50 are recommended.

Response:
We agree that the sample size is limited, as expected for a pilot study. The primary aim of this investigation was to explore temporal patterns in angiogenic markers following DEM-TACE and to assess the feasibility of a multiplex assay for these purposes. We have now more explicitly acknowledged the statistical limitations of our findings in Section 4.8 (Methodological Considerations and Study Limitations). While correlations with clinical outcomes would indeed strengthen the study, these were beyond the scope of the current design but are planned for future research with expanded cohorts.

  1. Hypoxia inference and lack of tissue-level HIF/VEGF analysis

Comment: Hypoxia is inferred but not demonstrated; tumor biopsies are needed for HIF-1/VEGF expression.

Response:
We appreciate this important point. As the current study was based on plasma sampling only, we could not directly assess tissue-level hypoxia markers. However, we have now clarified in both the Methods and Discussion sections that HIF-1α and VEGF expression at the tissue level were not measured due to the non-invasive design of the study and lack of biopsy sampling. We have expanded the discussion (Section 4.6 and 4.8) to emphasize the value of integrating plasma and tissue data in future studies to establish the mechanistic link between hypoxia and angiogenic response.

  1. No functional validation of angiopoietin-2/HGF

Comment: No assays confirm that increased angiopoietin-2 and HGF levels correspond to functional angiogenic activity.

Response:
Thank you for highlighting this. We agree that functional validation would be highly informative. However, the study was observational and plasma-based in nature, without access to tissue or functional in vitro assays. We have now noted this limitation in Section 4.8 and highlighted it as an important area for future research, potentially involving endothelial cell assays or co-culture models to confirm the biological relevance of circulating markers.

  1. Lack of integration with HIF pathway in discussion

Comment: The discussion lacks sufficient integration of the HIF pathway, a key driver of post-TACE angiogenesis.

Response:
We fully agree. We have now expanded Section 4.6 to include a more detailed discussion of the HIF pathway, its central role in hypoxia-induced angiogenesis, and how our observed biomarker patterns may reflect this axis. We also explain the limitations of plasma-based HIF-1α detection and propose future directions involving both plasma and tissue data integration.

  1. Concerns about low detectability of key markers (FGF-1/2, VEGF-A)

Comment: Key angiogenic markers were often undetected, raising concerns about assay sensitivity.

Response:
This is an important point. We have now emphasized in Section 3.4 (Results) and 4.8 (Discussion) that low detection rates of some factors were likely due to the lower sensitivity thresholds of the multiplex platform used. We suggest that future studies employ ultrasensitive detection platforms (e.g., single-molecule array [Simoa]) or high-sensitivity ELISA kits for better detection of low-abundance factors. This limitation has been more clearly noted in the revised manuscript.

Minor Comments

  1. Figure 2 lacks error bars

Comment: Figure 2 does not display variability.

Response:
We thank the reviewer for this observation. We have now revised Figure 2 to include appropriate error bars (standard deviation) to better reflect inter-individual variability in relative biomarker kinetics.

  1. Add recent background references on HCC

Comment: Add updated therapeutic/research references in the Introduction. Suggest PMID: 38223688 and PMID: 39521704.

Response:
We have now incorporated both suggested references in the Introduction to enrich the background discussion on recent advances in HCC treatment and biomarker research. Thank you for the useful recommendation.

  1. One-month sampling may miss late-phase angiogenesis

Comment: A 1-month endpoint may miss angiogenesis that peaks at 2–3 months post-TACE.

Response:
We appreciate this insight and agree. In Section 4.8, we now note that a longer follow-up sampling window (e.g., at 2–3 months) would be valuable for capturing late-phase angiogenic responses. This is a clear direction for future longitudinal studies.